# MDPE: A Multimodal Deception Dataset with Personality and Emotional Characteristics

## Abstract

Deception detection has garnered increasing attention in recent years due to the significant growth of digital media and heightened ethical and security concerns. It has been extensively studied using multimodal methods, including video, audio, and text. In addition, individual differences in deception production and detection are believed to play a crucial role.Although some studies have utilized individual information such as personality traits to enhance the performance of deception detection, current systems remain limited, partly due to a lack of sufficient datasets for evaluating performance. To address this issue, we introduce a multimodal deception dataset MDPE. Besides deception features, this dataset also includes individual differences information in personality and emotional expression characteristics. It can explore the impact of individual differences on deception behavior. It comprises over 104 hours of deception and emotional videos from 193 subjects. Furthermore, we conducted numerous experiments to provide valuable insights for future deception detection research. MDPE not only supports deception detection, but also provides conditions for tasks such as personality recognition and emotion recognition, and can even study the relationships between them. We believe that MDPE will become a valuable resource for promoting research in the field of affective computing.

## 1 Introduction

Generally, deception refers to the act of misleading, tricking, or deceiving others DePaulo et al. (2003). It involves hiding the truth or presenting false information to create an impression that is not accurate. Deception can take many forms, including both verbal and nonverbal information Burgoon et al. (2021). And it also occurs in various contexts, such as interpersonal relationships, business, politics, and entertainment. Deception is often considered unethical and can have serious consequences for trust and relationships.

As deception has expanded to other fields such as social media, interviews, online transactions, and deception in daily life,the need arises for a reliable and efficient system to aid the task of detecting deceptive behavior. Many machine learning approaches have been proposed in order to improve the reliability of deception detection systems Granhag & Hartwig (2008). In particular, physiological, psychological, visual, linguistic, acoustic, and thermal modalities have been analyzed in order to detect discriminative features and clues to identify deceptive behavior Feng et al. (2012); Hirschberg et al. (2005); Newman et al. (2003); Rajoub & Zwiggelaar (2014). Video-based deception detection is a current priority in deception research, because behavioral cues can be extracted from videos in a cheaper, faster, and non-invasive manner Burzo et al. (2018), which is preferable to invasive approaches that extract clues through devices attached to human bodies (e.g., polygraphs). Visual clues of deception include facial emotions, expression intensity, hands and body movements, and microexpressions. These features were shown to be capable of discriminating between deceptive and truthful behavior Ekman (2009); Owayjan et al. (2012). Acoustic features took into account the pitch and speaking rate among other measurements to specify whether certain features are associated with an act of deceit Hirschberg et al. (2005). Linguistic features were usually extracted from the language, words usage, and consistency of the statements made by a person Howard & Kirchhübel (2011). Recently, multimodal analysis has gained a lot of attention due to their superior performance compared to the use of unimodal modalities. In the deception detection field, several multimodal approaches Pérez-Rosas et al. (2015); Krishnamurthy et al. (2018); Şen et al. (2020); Mathur &

Matarić (2020) have been suggested to improve deception detection by integrating features from different modalities. This integration created a more reliable system that is not susceptible to factors affecting sole modalities and polygraph tests.

In addition, it is firmly believed that there are individual differences in deception production and detection Levitan et al. (2015); Majumder et al. (2017); Ren et al. (2021). Specifically, it includes cognitive level, personality traits, psychological characteristics, and emotional expression. Everyone has different personalities and psychological characteristics, and the expression of emotions is also various. It has been demonstrated through several studies that personality factors and emotional cues play a significant role in subjects' ability to deceive and detect deception Levitan et al. (2015); Gaspar & Schweitzer (2013). Emotion is a fundamental aspect of human communication that interacts with cognition, guiding social behavior in both human-to-human interactions and human-computer interactions Gordon et al. (2016); Marchi et al. (2015). Emotional characteristics are important, because deceptive behavior can trigger emotional states, leading to behavioral changes that serve as deceptive clues Ekman (2009); Vrij (2008). However, it is difficult to directly improve the accuracy of deception detection using emotional features Hartwig & Bond Jr (2014). One of the reasons is that emotional expression is also a part of deception. It is usually difficult to detect whether a deceiver's emotional expression is genuine or disguised.

To address this issue, we propose a multimodal deception dataset MDPE. It not only collects subjects' deception information, but also personality information and emotional expression information. Each subject was required to conduct another emotional experiment in addition to engaging in deception, in order to obtain their true emotional expression. Although our research was conducted in the laboratory to provide clear and comparable conversations, we provided subjects with effective monetary incentives to detect and generate effective deceptive behavior Levitan et al. (2015). To our knowledge, this is the largest multimodal deception dataset in the released dataset and the only deception detection dataset with personality and emotional characteristics.

To sum up, our contributions are threefold:

- We propose a novel multimodal deception dataset MDPE with personality and emotional characteristics, composed of facial video, and audio recordings and transcript. And an easily replicable experimental protocol has also been provided to researchers.
- We provide a benchmark for deception detection from multimodal signals, and discussed the impact of personality traits and emotional cues on deception detection.
- We offer new possibilities to facilitate further affective computing research, encourage the development of new methods that utilize individual differences for deception detection, as well as for tasks such as personality recognition and emotion recognition.

## 2 RELATED WORK

**Deception Dataset** Pérez-Rosas et al. Pérez-Rosas et al. (2015) introduced a new multi-modal deception dataset Real-life Trial having real-life videos of courtroom trials. They demonstrated the use of features from different modalities and the importance of each modality in detecting deception. They also evaluated the performance of humans in deception detection and compared it with their machine learning models. The Box-of-Lies dataset Soldner et al. (2019) was released with video and audio from a game show, and presents preliminary findings using linguistic, dialog, and visual features. Multiple modalities have been introduced in the hope of enabling more robust detection. Pérez-Rosas et al. Pérez-Rosas et al. (2014) introduced a dataset for deception including video and thermal imaging, as well as physiological and audio recordings. Gupta et al. Gupta et al. (2019) proposed Bag-of-Lies, a multimodal dataset with gaze data for detecting deception in casual settings. Speth Jeremy et al.Speth et al. (2021) proposed a multimodal deception database DDPM contains almost 13 hours of recordings of 70 subjects, as well as physiological signals such as thermal video frames and pulse oximeter data. Most studies on deception detection are designed and evaluated on private datasets, typically with relatively small sample sizes, and MDPE dataset addresses these drawbacks. Table 1 compares the sample size and length for existing datasets and MDPE.

**Multimodal Deception Detection** Decades of research in psychology, and deception detection have documented verbal and nonverbal behavioral cues indicative of deceptive communication. Visual cues

Table 1: Comparison of the subject count and length for several databases for deception detection

| dataset | Subjeet Count | Length(Minutes) |
| --- | --- | --- |
| Multimodal | 30 | - |
| Real Trials | 56 | 56 |
| Box-of-Lics | 26 | 144 |
| Bag-of-Lies | 35 | <241 |
| DDPM | 70 | 776 |
| MDPE | 193 | 6209 |

such as the frequency and duration of eye blinks Bhaskaran et al. (2011); Fukuda (2001); Minkov et al. (2012), dilation of pupils Dionisio et al. (2001); Lubow & Fein (1996), and facial muscle movements Hurley & Frank (2011); Porter et al. (2011) have been found to distinguish between deceptive and truthful behavior. Vocal cues can be indicative of deception, with deceptive speakers tending to speak with higher and more varied pitch DePaulo et al. (2003); Zuckerman et al. (1981), shorter utterances, and less fluency Rockwell et al. (1997); Sporer & Schwandt (2006) than truthful speakers. Deception also correlates with verbal attributes of speech, with deceivers tending to communicate with less cognitive complexity, fewer self-references, and more words indicative of negative emotions Zhou et al. (2004); Newman et al. (2003). Mohamed Abouelenien et al. Abouelenien et al. (2016) explored a multimodal deception detection approach and integrates multiple physiological, linguistic, and thermal features. They used a decision tree model, to gain insights into the features that are most effective in detecting deceit. Leena Mathur et al. Mathur & Matarić (2020) analyzed the discriminative power of features from visual, vocal, and verbal modalities affect for deception detection. They experimented with unimodal Support Vector Machines (SVM) and SVM-based multimodal fusion methods to identify effective features for detecting deception.

**Individual Difference Deception** Some studies confirm that some of the five NEO-FFI (Neuroticism-Extraversion-Openness Five-Factor Inventory) dimensions are related to deception Ramanaiah et al. (1994); Jakobwitz & Egan (2006). Sarah Ita Levitan et al. Levitan et al. (2015) reported the role of personality factors derived from the NEO-FFI and of gender, ethnicity and confidence ratings on subjects' ability to deceive and to detect deception. Justyna Sarzyńska et al. Sarzyńska et al. (2017) reports correlations between the ability to lie and extraversion, as well as conscientiousness. Personality characteristics are a promising set of information for deception detection, and similarly, emotional characteristics are also important. Joseph P. Gaspar et al. Gaspar et al. (2022) integrate prior theory and research on emotions, emotional intelligence, and deception and introduce a theoretical model. This model explores the interplay between emotional intelligence (the ability to perceive emotions, use emotions, understand emotions, and regulate emotions; and deception. Mircea Zloteanu et al hold strong beliefs about the role of emotional cues in detecting deception, and explored how decoders' emotion recognition ability and senders' emotions influence veracity judgements Zloteanu et al. (2021). Joseph P. Gaspar et al. Gaspar & Schweitzer (2013) believe that emotions are both an antecedent and a consequence of deception, and they introduce the emotion deception model to represent these relationships. This model broadens their understanding of deception in negotiations and accounts for the important role of emotions in the deception decision process. To our knowledge, MDPE is the only deception detection dataset with personality and emotional characteristics.

## 3 DATASET

### 3.1 MATERIALS

We collect our deception dataset using: a sports camera Gopro Hero9 with a resolution of 1920x1080 and a frame rate of 60 fps. The voices of the subjects are also recorded by the built-in microphone of the camera. A Thinkpad laptop was provided to subjects for watching emotion induction videos during the emotion experiment. The experimental place is in a professional recording stdio, and during the data collection process, only the subject and interviewer stay in the room. Some materials were prepared by the Data Collection Coordinator (DCC). The experimental setup is shown in Figure 1.

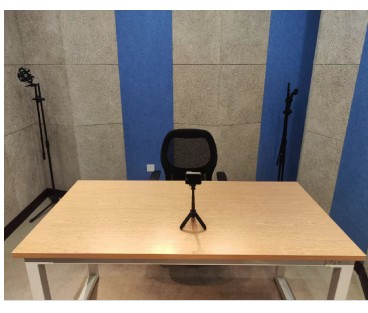 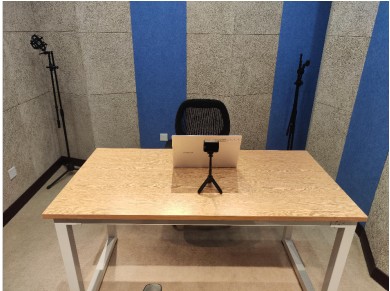

(a) Deception Experiment            (b) Emotional Experiment

Figure 1: The example of the place setup for data acquisition.

## 3.2 PARTICIPANTS

There were 193 subjects in this study, of which 130 were female and 63 were male. They are all native Chinese speakers from different backgrounds. Their age ranged from 18 to 69 years old, and they had various professions including students, workers, teachers, retirees, etc.

Firstly, we segmented the raw video, resulting in 1808 minutes of deceptive video and 4401 minutes of emotional video, totaling 6209 minutes. Each of the 193 subjects provided 24 responses, 9 of which were deceptive. The length of deception videos ranged from 4 minutes to 27 minutes. Each subject had 16 emotional videos, with lengths ranging from 19 minutes to 38 minutes (including the time spent watching emotional induction videos).

Before the experiment began, the subjects were informed of all experimental procedures. The subjects explicitly consented to record their conversation and publish the video data in a scientific conference or journal. And we do not publish any privacy sensitive data, and the anonymity of participants will be guaranteed. All data were collected under a protocol approved by the authors'institution's Human Subjects Institutional Review Board.

## 3.3 PROCEDURE

**Personality Characteristics Collection**: Subjects were required to fill out a Big Five personality questionnaire Zhang et al. (2022), which consists of 60 questions. Each question was marked with a score indicating whether the descriptions match their own, with 1 indicating strong disagreement and 5 indicating strong agreement. Details can be found in the appendix A.

**Emotional Experiment**: Subjects were asked to watch 16 emotional induction videos, including two induction videos for each emotion of sadness, happiness, relaxation, surprise, fear, disgust, anger, and neutral. There are a total of 39 induction videos, of which 17 are from the Chinese Emotional Video System (CEVS) et al. (2010). Each video segment has been labeled and evaluated to ensure that it can induce corresponding emotions. Another 22 are from our online collection. Because the CEVS only include six emotions: sadness, happiness, fear, disgust, anger, and neutral, and some videos of the CEVS are outdated and cannot successfully induce corresponding emotions in our pre-experiments. Each video we collected online was annotated by 12 data annotators based on CEVS selection criteria and evaluation methods, and the results showed that each video triggered strong emotions.

Before the emotional experiment began, the DCC randomly selected 16 induction videos (ensure two videos for each emotion) for the subjects to watch. After watching each video, subjects were required to describe their feelings and then fill out an "Emotional Scale", which quantified 8 emotions. Subjects rated their 8 emotions, ranging from 1 to 5, with 1 indicating no such emotion and 5 indicating the strongest emotion. Details of the emotion scale can be found in the appendix B.

**Deception Experiment**: The deception data collection process follows DDPM Speth et al. (2021). The interviewer conducted an interview with the subject, asking a total of 24 questions. These 24 questions were jointly formulated by 5 psychology researchers with over 5 years of experience. Before the interview, details of the emotion scale can be found in the appendix C. The DDC randomly

selected 9 questions that must lie and hand them over to the subject (the interviewer does not know which 9 questions). The first 3 questions will not be selected, which means that the first three "warm up" questions were always to be answered honestly. They allowed the subject to get settled, and gave the interviewer an idea of the subject's demeanor when answering a question honestly.

The subject have a maximum of 15 minutes to prepare, and during the preparation process, they must remember these 9 questions and think about how to deceive in the upcoming interview process. During the interview process, when asked these 9 questions, the subject must lie, and when asked the remaining 15 questions, they must tell the truth. Subjects were motivated to deceive successfully through two levels of bonus compensation: if they were able to deceive the interviewer in five or six of the nine deceptive responses, they were given a 150 percent of a base incentive payment; the base payment was doubled if they were successfully deceptive in seven or more questions. In order to collect more indistinguishable deception answers, we encourage subjects to incorporate some truth into lies when answering these deceptive questions.

During the interview process, the interviewer asked 24 questions in random order, and provide their judgment of truthful or deceptive answers to each question. And the interviewer filled out the "Interviewer Judgment Scale", which record the trust level the interviewer thinks of each answer. The trust level was divided into 1-5 points, where 1 represents definitely true and 5 represents definitely false. After the interview, the subject also filled out the "Subject Lie Confidence Scale" and be asked to rate the answer they just lied to. The same score is 1-5, where 1 represents that I have definitely deceived successfully and 5 represents that I have definitely not deceived successfully.

Each subject was required to complete the above three experiments, so that we can obtain their personality , emotional and deceptive characteristics.

## 4 BENCHMARK

### 4.1 DATA PREPROCESSING

For the visual modality, we first unify the raw video to 30 fps of the frame rate, and crop and align faces via DLib Toolkit King (2009). Figure 2 shows some examples of real and deceptive faces. Then, we use visual encoders to extract frame-level , followed by average pooling to compress them to the video level. For the audio modality, we use FFmpeg to separate the audio from the video and unify the audio format to 16kHz and mono. For the textual modality, we first extract transcript using Paraformer Gao et al. (2022), an open-source ASR toolkit. After data preprocessing, for each sample $x_i$, we extract acoustic features $f_i^a \in \mathbb{R}^{d_a}$, textual features $f_i^l \in \mathbb{R}^{d_l}$, and visual features $f_i^v \in \mathbb{R}^{d_v}$, where $\{d_m\}_{m \in \{a,l,v\}}$ is the feature dimension for each modality.

### 4.2 FEATURE EXTRACTION

Different features lead to distinct results. To guide feature selection, we evaluate the performance of different features under the same experimental setup.

**Visual Modality**: Compared with handcrafted features, deep features extracted from supervised models are useful for facial expression recognition Li & Deng (2020). CLIP Radford et al. (2021) is a multimodal model based on contrastive learning, where training utilizes text and images to construct positive and negative sample pairs. Pre-training is conducted on a dataset comprising 400 million pairs, resulting in strong generalization capabilities. And Vision Transformer (ViT) Dosovitskiy et al. (2020) is a transformer encoder model , pre-trained in a supervised manner on a large dataset of images. Images are presented to the model in the form of sequences of fixed-size patches (resolution of 16x16) and undergo linear embedding.

**Acoustic Modality**: We extracted the handcrafted feature extended Geneva Minimalistic Acoustic Parameter Set (eGeMAPS) Eyben et al. (2015), which contains 88 acoustic parameters designed specially for speech emotional recognition tasks, covering spectral, cepstral, and prosodic features . And Wav2vec Baevski et al. (2020) demonstrates the power of learning robust representations solely from speech audio, fine-tuning on transcribed speech, surpassing the best semi-supervised methods. It masks speech inputs in the latent space and addresses a contrastive task defined on quantized latent representations. It has been widely applied to downstream speech tasks. HUBERT Hsu et al.

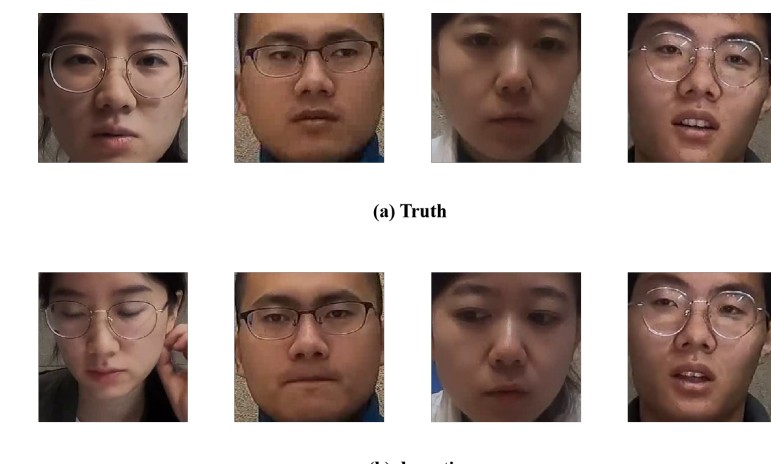

(a) Truth

(b) deception

Figure 2: Some examples of truth and deception faces.

(2021) utilizes offline clustering steps to provide aligned target labels for prediction losses. A key component of this approach is applying prediction losses only in masked regions, forcing the model to learn representations that combine acoustic and language models on contiguous inputs. To better distinguish between speakers, a sentence mixture training strategy WavLM Chen et al. (2022) is proposed, allowing for the unsupervised creation and merging of additional overlapping sentences during the training process.

**Text Modality**: Among all language models, BERT Devlin et al. (2018) and its variants are widely utilized, and it uses the masked language model and next sentence prediction objectives to learn word embeddings. Here, we extract the sbert-chinese-general-v2 features, which is based on the bert-base-chinese version of the BERT model and is trained on the million-level semantic similarity data set SimCLUE. ChatGLM Du et al. (2021) is an open-source conversational language model supporting bilingual question answering in both Chinese and English, based on the General Language Model (GLM) architecture. It demonstrats exceptional contextual understanding and more efficient inference capabilities. Baichuan Yang et al. (2023) is an open-source large-scale model with 13 billion parameters. It features a larger size, more extensive training data, and more efficient inference capabilities.

### 4.3 MODEL STRUCTURE

For unimodal features, we utilize the fully-connected layers to extract hidden representations and predict deception:

$$h_i^m = \text{ReLU}\left(f_i^m W_m^h + b_m^h\right), m \in \{a, l, v\} \tag{1}$$

$$\hat{y}_i = \text{softmax}\left(h_i^m W_m^d + b_m^d\right), m \in \{a, l, v\} \tag{2}$$

where $h_i^m \in \mathbb{R}^h$ is the hidden feature for each modality, $d_i \in \mathbb{R}^2$ is the estimated deception probabilities. For multimodal features, different modalities contribute differently to deception detection. Therefore, we compute importance scores $\alpha_i \in \mathbb{R}^{3 \times 1}$ for each modality and exploit weighted fusion to obtain multimodal features:

$$h_i = \text{Concat}\left(h_i^a, h_i^l, h_i^v\right) \tag{3}$$

$$\alpha_i = \text{softmax}\left(h_i^T W_\alpha + b_\alpha\right) \tag{4}$$

Table 2: Unimodal results on MDPE. "P" denotes the addition of personality features,"E" denotes the addition of emotional features

| Feature | Accuracy | AUC | with P | | with E | | with P and E | |
|---|---|---|---|---|---|---|---|---|
| | | | Accuracy | AUC | Accuracy | AUC | Accuracy | AUC |
| VIT | 60.30% | 0.577 | 61.27% | 0.582 | 61.43% | 0.615 | 61.55% | 0.620 |
| CLIP-base | 58.54% | 0.602 | 59.17% | 0.611 | 58.32% | 0.606 | 59.11% | 0.605 |
| CLIP-large | 57.30% | 0.574 | 58.34% | 0.582 | 56.97% | 0.555 | 57.67% | 0.579 |
| eGeMAPS | 55.86% | 0.563 | 57.22% | 0.588 | 56.22% | 0.577 | 56.89% | 0.585 |
| HUBERT-base | 58.13% | 0.615 | 62.38% | 0.651 | 59.35% | 0.617 | 62.12% | 0.641 |
| HUBERT-large | 60.80% | 0.636 | 62.07% | 0.646 | 60.34% | 0.621 | 61.87% | 0.644 |
| Wav2vec2-base | 58.75% | 0.581 | 59.74% | 0.603 | 59.99% | 0.594 | 59.84% | 0.599 |
| Wav2vec2-large | 60.10% | 0.582 | 61.88% | 0.617 | 59.32% | 0.592 | 62.10% | 0.634 |
| WavLM-base | 61.66% | 0.609 | 60.82% | 0.606 | 60.16% | 0.595 | 60.92% | 0.593 |
| WavLM-large | 57.82% | 0.599 | 60.31% | 0.583 | 58.02% | 0.607 | 60.52% | 0.611 |
| Sentence-BERT | 61.76% | 0.639 | 62.34% | 0.651 | 63.21% | 0.641 | 63.34% | 0.657 |
| ChatGLM2-6B | 60.73% | 0.648 | 61.45% | 0.659 | 61.45% | 0.667 | 61.56% | 0.676 |
| Baichuan-13B | **61.87%** | **0.649** | **62.90%** | **0.667** | **63.32%** | **0.675** | **63.74%** | **0.683** |

## 4.4 FEATURES FUSION

For personality and emotional expression features, we use concatenation for feature fusion. To represent personality traits, we utilize the scores derived from established personality scales, which provide a quantitative measure of individual characteristics. These scores serve as our primary personality features, capturing a broad spectrum of traits such as openness, conscientiousness, extraversion, agreeableness, and neuroticism.

For the emotional expression features, our process begins with training a dedicated emotion recognition model. This model is designed to analyze and interpret various emotional cues present in the input samples. We feed a comprehensive set of emotional expression features into this model, allowing it to learn and adapt to the nuances of emotional communication. Once the model has been trained, we extract features from the last fully connected layer, which encapsulates the learned representations of emotional expressions. To refine these features further, we apply average pooling, which helps in summarizing the information across different samples, yielding a robust representation of emotional expression. The resulting concatenated feature set, combining both personality and emotional expression elements, offers a rich and multidimensional view of individual behavior and emotional states.

## 4.5 IMPLEMENTATION DETAILS

We select the dimension of latent representations from $\{64, 128, 256\}$. During training, we use the Adam Kingma & Ba (2014) optimizer and choose the learning rate from $\{10^{-3}, 10^{-4}\}$. We set the maximum number of epochs to 300 and the weight decay to $10^{-5}$. Dropout Srivastava et al. (2014) is also employed, and we select the rate from $\{0.2, 0.3, 0.4, 0.5\}$ to alleviate the over-fitting problem. We randomly select 5 answers (3 truths and 2 deceptions) from 24 answers in all samples as the validation set, and the remaining 19 answers as the training set. To mitigate randomness, we run each experiment five times and report the average result. And we choose the cross-entropy as loss function and the accuracy and AUC as the evaluation metric.

## 4.6 EXPERIMENT RESULTS

**Unimodal Results** In this section, we establish the unimodal benchmark for MDPE and report results in Table 2. We hope this benchmark can provide guidance for feature selection and point the way to developing powerful feature extractors. For the visual modalities, VIT achieves better results than CLIP, possibly because VIT is trained on supervised datasets and can reveal more deceptive clues than CLIP features that use text as a supervisory signal. For the acoustic modality, the deep features outperform handcrafted features. This may be because egemas are features related to emotions, using a emotion-related handcrafted acoustic feature may limit performance. In contrast, deep features can capture more universal acoustic representations for deception detection. For the textual modality,

Table 3: Multimodal results on MDPE. We select several well-performing unimodal features and report their fusion results. Here, "V", "A" and "T" represent the visual, acoustic and textual modalities, respectively.

| V | A | T | Accuracy | AUC | with P | | with E | | with P And E | |
|---|---|---|---|---|---|---|---|---|---|---|
| | | | | | Accuracy | AUC | Accuracy | AUC | Accuracy | AUC |
| VIT | HBB | - | 61.76% | 0.628 | 61.53% | 0.617 | 61.14% | 0.616 | 62.68% | 0.628 |
| VIT | WMB | - | 60.88% | 0.611 | 60.22% | 0.599 | 60.02% | 0.599 | 60.72% | 0.587 |
| CLB | HBB | - | 61.02% | 0.618 | 60.14% | 0.606 | 60.53% | 0.614 | 61.45% | 0.625 |
| VIT | - | Bai | 63.31% | 0.664 | 63.47% | 0.678 | 63.31% | 0.676 | 63.48% | 0.672 |
| CLB | - | Bai | 63.31% | 0.666 | 63.52% | 0.677 | 63.38% | 0.667 | 63.10% | 0.664 |
| CLL | - | Bai | 64.15% | 0.665 | 63.94% | 0.679 | 63.98% | 0.672 | 64.04% | 0.678 |
| - | HBL | Bai | 62.69% | 0.658 | 63.42% | 0.665 | 63.00% | 0.665 | 63.42% | 0.663 |
| - | W2B | Bai | 63.83% | 0.663 | 63.48% | 0.663 | 63.57% | 0.668 | 63.79% | 0.664 |
| - | WMB | Bai | 64.25% | 0.661 | 64.15% | 0.679 | 63.90% | 0.677 | 64.07% | 0.679 |
| VIT | HBB | Bai | **64.45%** | **0.675** | 64.33% | 0.679 | 63.93% | 0.674 | 64.00% | 0.675 |
| VIT | WMB | Bai | 63.42% | 0.666 | **64.87%** | **0.681** | 63.62% | 0.664 | 63.59% | 0.672 |
| CLB | HBB | Bai | 62.94% | 0.657 | 63.93% | 0.678 | **63.97%** | **0.678** | **64.66%** | **0.687** |

we focus on textual encoders that support Chinese (large language models are generally trained on multilingual corpora containing Chinese). And the textual modality can achieve best performance than the visual and acoustic modalities, which indicates that our dataset textual reveals more deceptive clues than other modalities.

Of course, we have also incorporated personality and emotional features into deception detection. Models trained with personality features almost often exhibit superior performance. These findings demonstrate the importance of personality in deception detection tasks. And the addition of emotional features has also improved the performance of deception detection models, but the improvement is not comparable to personality features, even some results have decreased. This may be because personality traits are directly usable features, and emotional features are extracted by emotion recognition models. The quality of features is also influenced by the emotion recognition models. In the future, better methods for using emotional expression features can be explored. By incorporating both personality and emotional features, the deception detection model achieved the highest performance in unimodal results, These results show that personality and emotional expression characteristics are indeed important for deception detection tasks, and using them can truly achieve deception detection modeling based on individual differences. It is worth mentioning that among all the unimodal results, Baichuan always achieved the best results. It is the model with the largest number of parameters among the features we used, and it further demonstrates the potential of large language models in deception detection.

**Multimodal Results** In Table 3, we select several well-performing unimodal features and report their fusion results. Experimental results demonstrate that multimodal fusion consistently improves performance. The reason lies in the fact that deception cues can be conveyed through multiple modalities. The integration of multimodal information allows the model to better comprehend the video content and accurately detection deception. Firstly, almost all features have achieved performance improvements in multimodal feature fusion, but the fusion of visual and acoustic modalities has hardly improved or even declined. It indicates that the addition of text features make the performance of the model tend to stabilize. This is consistent with human judgment. When people judge whether others are lying, they tend to express the truth or falsehood of the content, because it difficult to judge from visual or acoustic features. In the future, further exploration and research should be conducted on deceptive clues in visual and acoustic features.

Similar to unimodal results, models trained with personality features typically exhibit excellent performance. The addition of emotional features makes the performance of the model unstable. Some features have been improved, while others have decreased. Finally, the best result always comes from the three modalities fusion, and the three modalities fusion with personality and emotional characteristics has achieved the best performance.

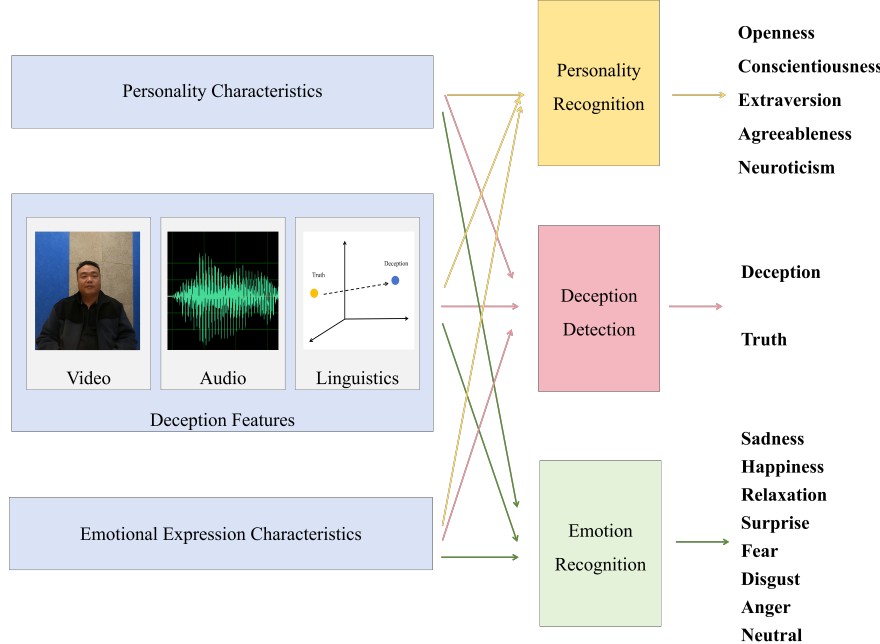

Figure 3: Possible future work on the MDPE

## 5 DISCUSSION

### 5.1 FUTURE WORK

Firstly, dimensional emotions are also important. In fact, we are labeling MDPE with dimensional emotions (including deception and emotion), which can not only study the impact of dimensional emotional features on lie detection at the individual level, but also explore more emotional clues in the deception process. Secondly, we extracts deep features from some pre-trained models and uses simple models for deception detection. In the future, larger models should be designed to be used for deception detection tasks. Thirdly, this article simply concatenates personality and emotional expression features to assist in deception detection tasks. In the future, more complex feature fusion or model fusion algorithms can be used for deception detection. Finally, in this paper, we only focuses on deception detection tasks, but MDPE includes individual personality, deception, and emotional expression information, which can not only support deception detection tasks, but also tasks such as personality recognition and emotion recognition. Even these individual level information can be used to assist other tasks, as shown in Figure 3. In fact, there have been many studies on the relationship between personality and emotion Hughes et al. (2020); Li et al. (2022); Zhang et al. (2019), but the lack of relevant datasets has led to slow research progress in this field, and we believe that MDPE can provide valuable resources for these research directions.

### 5.2 LIMITATIONS

Firstly, although the subjects were required that they must lie about the deception questions, and verified the deceptive questions and content with the Interviewer after the deception experiment, we do not know whether the subjects have actually deceived on the deception questions. Secondly, although our induction videos have been annotated by professionals to demonstrate their reliability and validity, there are still some subject whose feelings are inconsistent with the emotions we expected to induce. This is because different people have different understandings of the video content, triggering different emotions. Thirdly, relying on self-assessment scales for data annotation is a subjective process for subjects, which may lead to bias in subsequent analysis. Different subjects may have significant differences in their perception of emotions. In addition, MDPE only collects native Chinese speakers, there may be cultural differences in deception detection. Finally, gender

imbalance among subjects in MPDE is a common issue in human data collection D'Mello et al. (2022); Pinho-Gomes et al. (2022).

## 6 CONCLUSION

We present MDPE, a dataset for deception detection featuring three categories of data modalities, as well as personality and emotional characteristics. Firstly, diverse modalities carry complementary information that can be jointly exploited. By providing access to multiple synchronized modalities, MDPE enables cross-modal analyses that have the potential to improve the understanding of the relationships between video, audio, and text. Secondly, it can help improve the understanding of deceive behavior, aiming to develop reliable deception detection algorithms and enhance the security issues related to deception in our society. Thirdly, MDPE provides the personality traits and emotional expression characteristics of each subject, which can help analyze the impact of personality and emotional expression on deceive behavior. In addition, MDPE not only supports deception detection models, but also provides conditions for personality recognition and emotion recognition tasks, and can even study the relationship between deception, personality, and emotion, such as using personality features to improve the performance of emotion recognition tasks. Finally, to promote reproducibility, MDPE also provided a set of benchmark experiments. Although the proposed model focuses on deception detection, the use of personality and emotional features also demonstrates the predictive potential of our dataset. They represent a good starting point for future work that researchers and developers can use as a benchmark. By openly sharing MDPE, we hope to ignite new advances in this critical area of affective computing.

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

## A    APPENDIX

### A.1    BIG FIVE PERSONALITY INVENTORY SECOND EDITION (BFI-2)

Below are some descriptions of personal characteristics, some may or may not apply to you. Please fill in the corresponding number on the horizontal line before each sentence below to indicate whether you agree or disagree with this description.

1. Outgoing personality, enjoys socializing
2. Soft hearted and compassionate
3. Lack of organization
4. Calm and adept at handling pressure
5. Not very interested in art
6. Strong and confident personality, daring to express one's own opinions
7. Humble and respectful towards others
8. Relatively lazy
9. Being able to maintain a positive attitude even after experiencing setbacks
10. Interested in many different things
11. I rarely feel excited or particularly want to do anything
12. Often picking on others' faults
13. Reliable and reliable
14. Irregular mood and frequent emotional fluctuations
15. Skilled in creativity and able to find smart ways to do things
16. Relatively quiet
17. Lack of empathy towards others
18. Work in a planned and organized manner
19. Easy to get nervous
20. Enthusiastic with art, music, or literature
21. Often in a dominant position, like a leader
22. Often having disagreements with others
23. It's difficult to start taking action to complete a task

24. Feeling secure and satisfied with oneself

25. Disliking discussions with strong knowledge or philosophy

26. Not as energetic as others

27. Be magnanimous and magnanimous

28. Sometimes I lack a sense of responsibility

29. Emotionally stable and less likely to get angry

30. Almost no creativity

31. Sometimes shy and introverted

32. Helpful and selfless towards others

33. Habit keeps things tidy and orderly

34. Often worried and worried about many things

35. Valuing Art and Aesthetics

36. Feeling difficult to influence others

37. Sometimes being rude to people

38. Efficiency, starting and ending with work

39. Often feeling sad

40. Deep thinking

41. Full of energy

42. Do not trust others and doubt their intentions

43. Reliable, always trustworthy to others

44. Able to control one's emotions

45. Lack of imagination

46. Loud and talkative

47. Sometimes cold and indifferent to others

48. It's messy and doesn't like to tidy up

49. Rarely feel anxious or afraid

50. Feeling bored with poetry and drama

51. I prefer to have others take the lead and take responsibility

52. Humility and courtesy towards others

53. Have perseverance and be able to persist in completing tasks

54. Often feeling depressed and unhappy

55. Not very interested in abstract concepts and ideas

56. Full of enthusiasm

57. Think about people in the best possible way

58. Sometimes they may engage in irresponsible behavior

59. Emotions are variable and prone to anger

60. Creative and able to come up with new ideas

## A.2 EMOTIONAL SACLE

After watching the video, you need to rate the following emotions: sadness, relaxation, happiness, surprise, fear, anger, disgust, and neutral. Mark to what extent you feel it appropriately expresses your feelings, with intensity ranging from 1 to 5, where 1 is the least intense and 5 is the strongest.

| Video Number | Sadness | Relax | Happiness | Surprise | Fear | Angry | Disgust | Neutral |
|---|---|---|---|---|---|---|---|---|
| 1 | | | | | | | | |
| 2 | | | | | | | | |
| 3 | | | | | | | | |
| 4 | | | | | | | | |
| 5 | | | | | | | | |
| 6 | | | | | | | | |
| 7 | | | | | | | | |
| 8 | | | | | | | | |
| 9 | | | | | | | | |
| 10 | | | | | | | | |
| 11 | | | | | | | | |
| 12 | | | | | | | | |
| 13 | | | | | | | | |
| 14 | | | | | | | | |
| 15 | | | | | | | | |
| 16 | | | | | | | | |

### A.3 INTERVIEW QUESTIONS

1. What color do you like the most? Why?
2. Where is your hometown? Please briefly introduce it.
3. Do you have any hobbies?
4. Have you traveled in the past year?
5. How do you like Beijing?
6. What is your happiest experience?
7. What is your favorite food?
8. What is your personality like?
9. What is your biggest weakness?
10. What is your greatest strength?
11. What do you usually do to relax?
12. Which exercise or sport do you like?
13. Briefly introduce your family members.
14. Who is the person you have the greatest influence on you?
15. Do you have any special places or tourist destinations you want to go to?
16. Who is your favorite celebrity or great person?
17. What is your opinion on the words "neijuan" and "tangping"?
18. What is your favorite literary and artistic work?
19. Have you ever received any rewards or honors in school or at work?
20. What was your most unforgettable experience in the past year?
21. Have you participated in any major event?
22. Have you ever cheated in school or work?
23. Have you concealed a fact to your family or friends in the past year?
24. Have you ever lied to avoid responsibility?

