# OpenReview forum: "MDPE: A Multimodal Deception Dataset with Personality and Emotional Characteristics"
_ICLR.cc/2025/Conference — Submitted to ICLR 2025_

### Official Review · Reviewer_Jk8p · 2024-10-27

**Soundness:** 2
**Presentation:** 2
**Contribution:** 3
**Rating:** 3
**Confidence:** 3

**Summary:**

The manuscript introduces the MDPE dataset, a multimodal resource for contributing to deception detection research. This dataset is organized by more than 104 hours of video, audio, and text recordings from 193 subjects, along with annotations on personality traits and emotional expressions. The authors argued that previous deception detection datasets were limited in scope, often lacking the inclusion of individual characteristics that could enhance deception prediction. The authors also demonstrated the effectiveness of MDPE via a series of experiments, indicating that multimodal and personality-aimed models perform better than unimodal models. The manuscript concludes by emphasizing the potentiality of the dataset for future research with consideration of three key tasks, deception detection, personality recognition, and emotion recognition.

**Strengths:**

S1: The introduced dataset can be one of the good contributions for deception research, providing diverse modalities with consideration of personality and emotion annotations.
S2: The dataset's size is valuable, compared to other datasets. The authors provide the comprehensive dataset of 193 subjects with more than 104 hours. It represents a variety of demographics, covering a broad range of emotions and incorporating deception and genuine responses.
S3. The authors perform extensive benchmarking with multiple features (e.g., visual, acoustic, textual) and combine them with personality and emotional data, leading to valuable findings about multimodal fusion effectiveness.

**Weaknesses:**

I think that the first weakness of the manuscript is that the collected dataset includes only Chinese speakers, which are limited in its generalization. As the authors know, how do we apply this approach to future research considering other cultures and languages? Obviously, their selected text module (with Chinese-oriented LLM) is effective for their collected dataset. However, it can be applied to other datasets?

Second, although the authors indicate their approaches, I cannot find any demographic statistics in their collected dataset. Is it well-distributed with consideration of age, gender, and a number of characteristics?

Third, the authors mentioned that emotional characteristics are included in the dataset, but the manuscript lacks a deep analysis of how these characteristics influence deception, instead focusing more on personality traits. This leads to underutilizing the dataset's emotional potential. Moreover, the authors are required to present whether the emotion recognitions are presented with multi-labels or not.

**Questions:**

1. How do we adapt MDPE to other languages and cultures? Would personality and emotion still be as relevant in deception detection across different demographics?

2. What kinds of multimodal modules can we consider for future research? As presented in the manuscript, the authors considered several previously presented modules, but there can be additional selections for each module.

3. Although the authors consider multi-task approaches for this topic, I think that it is required to compare the results with other emotion recognition or personality-detection models. How do the authors think?

---

> ### Author Response · Authors · 2024-11-19
>
> Thank you very much for your guidance and valuable suggestions. We are very grateful for your comments, which are very helpful for us to sort out the structure and motivation of the article.
> ﻿
> 1. We acknowledge that our dataset, which includes only Chinese speakers, may limit its generalizability to other cultures and languages. To adapt the MDPE dataset for future research in different linguistic and cultural contexts,  Future studies could involve collecting similar datasets from diverse cultural backgrounds, allowing for comparative analyses that explore how cultural factors influence deception detection. In fact, we have started collecting and processing an extended version of MDPE data, which will greatly alleviate bias issues caused by population and cultural differences.
> ﻿
> 2. In fact, we have a description of population statistics (section 3.2). In the updated manuscript, we provide a more detailed introduction to the detailed demographic distribution of participants, taking into account factors such as age, gender, and other relevant characteristics, in order to have a clearer understanding of the composition of the dataset.
> ﻿
> 3. We recognize the need for a deeper analysis of how emotional characteristics influence deception. In the revised manuscript, we will expand our discussion to include: A thorough examination of the emotional traits present in the dataset and their potential impact on deception detection. And emotion recognition is presented with multi-labels.
> ﻿
> 4. Regarding the relevance of personality and emotion in deception detection across different demographics, we believe that while the underlying psychological constructs may remain relevant, their expression and interpretation can vary  across cultures. We will address this point in the updated manuscript, emphasizing the importance of cultural considerations in future research.
> ﻿
> 5. We appreciate your suggestion to explore additional multimodal modules for future research. In the revised manuscript, we will include a discussion of potential multimodal approaches that could enhance our understanding of deception detection.
> ﻿
> 6. While the primary focus of this study is deception detection, we acknowledge the creativity in the proposed comparison with other emotion recognition or personality detection models. We believe such comparisons could serve to validate the performance of personality and emotional traits in our context. Consequently, we will include a comparative analysis section in the new manuscript, assessing the performance of our approach against existing models.
> ﻿
> ﻿
> We hope that these revisions address your concerns and strengthen the contribution of our paper. We are grateful for the opportunity to improve our work based on your valuable feedback. Thank you again for your comments and acceptance of our views.

---

### Official Review · Reviewer_ZBrA · 2024-10-31

**Soundness:** 2
**Presentation:** 1
**Contribution:** 1
**Rating:** 1
**Confidence:** 5

**Summary:**

The authors have introduced a multimodal deception dataset. The main novelty of this dataset is that it includes not only deception features but also individual differences in personality and emotional expression characteristics.

**Strengths:**

The topic of deception detection is a challenging and important one.

**Weaknesses:**

My main concerns with this paper are about the novelty, the content and the presentation style. Regarding the content, the contribution of this paper seems quite limited for a top conference on machine learning (alternative and potentially more adequate conferences could be ICMI. CSCW, ACM-MM, etc.). Regarding the presentation, there are several typos and some sections of methods (Section 4.2 for example) are very badly organized and written in a not adequate manner.

**Questions:**

The 24 questions used by the interviewer with a given subject were formulated specifically for this study?

---

> ### Author Response · Authors · 2024-11-18
>
> Thank you very much for your guidance and valuable suggestions. We are very grateful for your comments, which are very helpful for us to sort out the structure and motivation of the article.
>
> ﻿
> In this study, we introduce a comprehensive multimodal deception detection dataset that encompasses visual, audio, and textual data, along with personality and emotional characteristics. This dataset is the largest of its kind and offers a unique opportunity to explore how personality and emotional traits can augment deception detection. Our work not only lays the groundwork for future research in affective computing but also suggests potential avenues for improving emotion recognition through the integration of personality insights and enhancing personality recognition performance with emotional cues. We have conducted extensive experiments that preliminarily demonstrate the value of incorporating personality and emotional traits in deception detection tasks. While we acknowledge that our application of machine learning methods may not be as advanced as desired, we believe that deception detection is inherently challenging, and the performance of various machine learning approaches has been modest. This indicates a need for further in-depth feature exploration within this dataset by the research community. Consequently, we have chosen to present a straightforward machine learning approach as a baseline for our analysis.
>
> ﻿
> We apologize for the typos and organizational issues in the previous version of the manuscript. We have thoroughly proofread the paper and corrected all identified typos. Additionally, we have restructured Section 4.2 to improve clarity and readability. We believe these changes will make our methods more accessible and better understood by the readers.
> ﻿
>
> The 24 questions used by the interviewer for specific topics were specifically designed for this study (line 215), 5 psychology researchers mainly referred to the Fraud Triangle Theory and Rational Choice Theory. The question design integrated knowledge from psychology, criminology, and sociology to comprehensively capture multiple aspects of fraudulent behavior. We used the Delphi method to collect and integrate expert opinions, and refined the issues through focus group discussions. The design of some issues reflects consideration of the new trends in current fraudulent behavior, aiming to capture aspects that traditional methods may overlook. In fact, after designing the initial interview questions, we conducted pre experiments, collected feedback from interviewees, and revised the interview questions after further discussion, the final interview questions were actually the third version.
> ﻿
>
> We appreciate your suggestion of alternative conferences such as ICMI, CSCW, and ACM-MM. While we believe our work is still relevant for this conference, we will consider your suggestion for future submissions if our work is not accepted here.
> We hope that these revisions address your concerns and strengthen the contribution of our paper. We are grateful for the opportunity to improve our work based on your valuable feedback. Thank you again for your comments and acceptance of our views.

---

### Official Review · Reviewer_bdmX · 2024-11-01

**Soundness:** 2
**Presentation:** 2
**Contribution:** 3
**Rating:** 3
**Confidence:** 5

**Summary:**

In this study, the authors collected data by conducting extensive interviews to create a multimodal deception dataset featuring 193 participants. This dataset also includes information about the participants' personalities and emotional states. Furthermore, the authors explored how features from different modalities and labels impact and enhance the deception detection task.

**Strengths:**

The authors conducted the data collection with great effort and provided comprehensive coverage regarding the data collection and label design.

**Weaknesses:**

1. The 'Introduction' section is somewhat verbose. While the authors did an excellent job of providing detailed background information on deception with examples from various modalities, I believe that some of this content, particularly the details in the second and third paragraphs, would be better suited for the 'Related Work' section.

2. The authors mention "effective incentives" without clarifying what these entail (whether monetary or non-monetary) or how they were distributed. Additionally, there is a typo in Table 1.

3. I suggest that the authors include a statement at the beginning of section 3.3 to clarify the order of the data collection activities.

4. The abbreviation "DDC" at the end of page 4 is unclear—are you referring to "DCC" instead?

5. The role of the 'interviewer' is crucial in this data collection process, as they are responsible not only for asking questions but also for being deceived and labeling responses. However, the manuscript lacks details about the interviewer's background and responsibilities.

6. More specific information is needed regarding how the anonymity of participants is ensured, including what information has been removed or altered.

7. The statement, "some videos of the CEVS are outdated and cannot successfully induce corresponding emotions in our pre-experiments," is also unclear. What does "outdated" mean in this context, and why exactly are these videos unable to evoke specific emotions?

8. Additionally, what is the source of the 22 online collected videos, and how did the 12 annotators reach a consensus on the labels?

9. In section 4.4 (feature fusion), the second paragraph is poorly explained. After reading it, I'm still unclear about the structure of the "dedicated emotion recognition model" and what specific emotional cues it utilizes. Descriptions such as "We feed a comprehensive set of emotional expression features into this model, allowing it to learn and adapt to the nuances of emotional communication" are too abstract.

10. Similarly, in the results presented in Table 2 and in the discussions regarding personality and emotional features on pages 7, 8, and 9, there is a lack of detail about what these features are and how they contribute.

11. I would also like more information about the 24 interview questions (as detailed in Appendix A.3) that were carefully designed by experienced psychology researchers. Are there any references or theories they used to create these questions?

Overall, the writing can be significantly improved, as there are many uncertainties and unclear descriptions throughout the manuscript. For example: "Before the interview, details of the emotion scale can be found in Appendix C," and "Some studies confirm that some of the five NEO-FFI (Neuroticism Extraversion-Openness Five-Factor Inventory) dimensions are related to deception."

In conclusion, it is evident that the work feels rushed, and substantial revisions are necessary to address the missing details. As it stands, it cannot be accepted in its current form.

**Questions:**

Please see the questions raised under the 'Weaknesses' section.

---

> ### Author Response · Authors · 2024-11-19
>
> Thank you very much for your guidance and valuable suggestions. We are very grateful for your comments, which are very helpful for us to sort out the structure and motivation of the article.
>
> 1. In the revised manuscript, we have condensed the "Introduction" section to enhance conciseness and shifted detailed background information, particularly from the second and third paragraphs, to the "Related Work" section. This reorganization improves the flow and maintains focus within the introduction.
> 2. We have clarified that the "effective incentives" mentioned (line 76) are monetary in nature and have provided details on their allocation (line 224) to effectively motivate participants in deceptive behavior. Additionally, we have meticulously revised the manuscript to address all linguistic inconsistencies and spelling errors, including those in Table 1, ensuring the accuracy and integrity of the presented data.
> 3. At the commencement of section 3.3, we have included a statement delineating the sequence of data collection activities. Participants initially complete the Big Five personality questionnaire to ascertain personality traits. Subsequently, half of the participants engage in emotional experiments while the other half partake in deception detection experiments. This balanced approach to experimental sequencing is intended to mitigate any potential influence of the sequence on the outcomes.
> 4. We have corrected the abbreviation "DDC" to "DCC" and offered a clear definition to prevent any ambiguity.
> 5. Furthermore, we have expanded upon the roles of the interviewers, detailing their background and responsibilities throughout the data collection process, thereby underscoring their pivotal role in our research.
> 6.  To ensure participant anonymity, we have provided explicit details regarding the measures taken to remove or alter identifying information. The experimental videos provided for analysis exclude any segments containing personal details such as names, contact information, place of residence, education, occupation, and so forth.
> 7.  The statement regarding CEVS videos being "outdated" and unable to effectively induce corresponding emotions refers to content that is no longer relevant or relatable to contemporary viewers. For instance, some videos intended to elicit happiness contained jokes that required historical context unfamiliar to most participants. As a result, these videos were replaced.
> 8. The 22 online videos were selected and annotated by 12 data annotators based on the CEVS criteria and evaluation methods (line 205). For further details, please refer to [1].
> 9.  In section 4.4, we have revised the second paragraph to elucidate the structure of the specialized emotion recognition models and the specific emotional cues they utilize. We have also detailed the role of personality and emotional traits in fraud detection tasks.
> 10. The 24 interview questions were meticulously designed for this study (line 215), with input from five psychology researchers drawing primarily from the Fraud Triangle Theory and Rational Choice Theory. The questions were crafted to integrate knowledge from psychology, criminology, and sociology, aiming to comprehensively capture various facets of fraudulent behavior. The Delphi method was employed to gather and synthesize expert opinions, and the questions were refined through focus group discussions. Some questions were designed to reflect current trends in fraudulent behavior, capturing aspects that traditional methods might overlook. After an initial design phase, we conducted pre-experiments, collected feedback from participants, and revised the interview questions accordingly, resulting in the final set being the third iteration.
> 11. We have thoroughly revised the manuscript to enhance writing quality and address all language issues, including uncertainties and unclear descriptions as highlighted by your review.
> ﻿
>  We hope that these revisions address your concerns and strengthen the contribution of our paper. We are grateful for the opportunity to improve our work based on your valuable feedback. Thank you again for your comments and acceptance of our views.

---

> > ### Comment · Reviewer_bdmX · 2024-11-23
> > **Response to authors by reviewer bdmX**
> >
> > Dear Authors,
> >
> > Thank you for considering the concerns I have raised. Have you uploaded the revised version yet? I checked the uploaded PDF file, which still contains all the previous issues.
> >
> > If a revised version is available, please upload it.

---

### Official Review · Reviewer_QeLm · 2024-11-04

**Soundness:** 2
**Presentation:** 1
**Contribution:** 1
**Rating:** 3
**Confidence:** 4

**Summary:**

The paper presents a new dataset for deception detection and emotion recognition.
The authors evaluate several approaches/features on the task of deception detection, also including emotion information.

**Strengths:**

The new dataset can be valuable to the community if made publicly available.
The authors present extensive evaluations of existing methods/features on the dataset.

**Weaknesses:**

The concept of emotion and emotion expression used in the paper is fuzzy.
E.g. what is a "true emotional expression" (line 074) supposed to refer to? One possibility that I might suspect is that the authors refer to whether the emotional expression is aligned with an internal state. This is a highly complex topic as emotions are not displayed directly on e.g. a person's face but are subject to regulation processes and social display rules (e.g. see Schneeberger et al., 2023; Müller et al., 2024).


The method novelty is limited, as the authors only evaluate simple combinations of existing approaches. This is not a very significant issue, as the paper presents itself as a dataset paper, where method novelty is not necessary imo. However, in my understanding it would then be important for an ICLR paper to have a more in-depth analysis of what are the implications of the new dataset for deep learning. We should be able to learn something with it. E.g., what are the remaining challenges, does it tell us something about what the current architectures cannot cover?
On what kinds of examples do the current methods fail?


References are missing for the statement "Most studies on deception detection are designed and evaluated on private datasets, typically with relatively small sample sizes" (line 104).
It would also be helpful to add information on the public availability of the existing datasets in Table 1.

In the description of the emotional experiment (lines 198-207), authors mention that they extended some existing dataset of emotional videos because it only has 6 emotions. In the end there seem to be 8. Which emotions were added and why were they considered to be relevant in the context of deception detection?

Concerning the results - the gains by including personality or emotion into the models are only marginal. Are these gain statistically significant when accounting for randomness (of the ssample/datasets, and of the methods/random number generator intialization)?

Research on deception detection raises a host of ethical issues which are not discussed in the paper.

All in all, the strength of the contribution is not too convincing to me and there are a number of open questions on the motivation/scientific background. Furthermore, the quality of the writing is not good enough at the moment, and the improvements for personality/emotion integration are only marginal.


Misc:
-----

- line 015: "role.Although"
- line 093: "introduced a new multi-modal deception dataset" - the characterisation "new" does not make sense here, as the dataset is already quite old (2015) and of course everything was new when it was introduced.
- line 102: wrongly formatted reference ("Speth Jeremy et al.Speth et al. (2021)"), this occurs several times, e.g. in 126,129 and at many other places.
- line 110: "Subjeet", missing space in "Length(Minutes)"
- line 133: non-grammatical paragraph heading "Individual Difference Deception"
- there are more formal/languageissues throughout the paper. the ones above are just some examples.
- line 376: "egemas"


References:
-----------

Schneeberger, Tanja, et al. "The deep method: Towards computational modeling of the social emotion shame driven by theory, introspection, and social signals." IEEE Transactions on Affective Computing (2023).

Müller, Philipp, et al. "Recognizing Emotion Regulation Strategies from Human Behavior with Large Language Models." arXiv preprint arXiv:2408.04420 (2024).

**Questions:**

What was the reasoning behind using a movie-based emotion induction procedure in the emotion experiment? Why is this procedure relevant in relation to the deception experiment where emotions are induced in a social situation?

**Details Of Ethics Concerns:**

Deception detection might be used to make people more transparent despite their will.

---

> ### Author Response · Authors · 2024-11-18
>
> Thank you very much for your guidance and valuable suggestions. We are very grateful for your comments, which are very helpful for us to sort out the structure and motivation of the article.
> ﻿
> 1. The concept of 'true emotional expression' pertains to the congruence between an individual's outward emotional display and their internal affective state. Our emotion induction procedure establishes a baseline for emotional responses, which is crucial for comparison with the emotional responses observed during the deception experiment. Experimental results have indicated that emotional features significantly contribute to fraud detection tasks.
> 2. In this study, we present a comprehensive multimodal deception detection dataset, encompassing visual, audio, and textual data, as well as personality and emotional features. This dataset, being the largest of its kind, offers a unique opportunity to investigate how personality and emotional traits can enhance deception detection. Our work not only lays the foundation for future research in emotion computing but also suggests potential methods to improve emotion recognition by integrating personality insights and enhancing personality recognition performance with emotional cues. Through extensive experiments, we have preliminarily demonstrated the value of integrating personality and emotional traits in deception detection tasks. While our application of machine learning methods may not have achieved the desired level of advancement, we believe that deception detection is inherently challenging, with various machine learning methods showing limited performance. This suggests a need for the research community to further explore the features and methods within this dataset. Thus, we propose a simple machine learning method as a baseline for our analysis. In the revised manuscript, we have conducted a more in-depth analysis of the impact of machine learning on new datasets, including remaining challenges and insights into the limitations of current architectures, and how this dataset can contribute to advancing research in this field.
> 3. In the updated manuscript, we have included information on the public availability of existing datasets.
> 4. Building upon the six emotions covered by the Chinese Emotional Video System (CEVS)—happiness, sadness, anger, fear, disgust, and neutrality—we have added relaxation and surprise to our dataset. Relaxation serves as an emotional baseline, allowing for the comparison of other emotional states. Deviations from a relaxed state during specific questions or situations may indicate deception, as inconsistent relaxation levels could suggest an attempt to deceive. Surprise, a spontaneous and difficult-to-fake emotion, can indicate truthfulness or deception; a genuine surprise response to an unexpected question or accusation may suggest unpreparedness and honesty, while a lack of surprise may indicate a prepared deceptive response.
> 5. Our experiments were conducted at least five times with randomly selected samples, and the average results were taken to ensure statistical significance.
> 6. Regarding ethical issues, we ensured that all experimental procedures were explained to the subjects, who provided explicit consent for the recording and publication of their conversation and video data in scientific conferences or journals. Our data collection and dissemination adhere to the principle of informed consent and comply with relevant laws, regulations, and ethical review requirements, all approved by our institution's Human Subjects Institutional Review Board. We have implemented privacy protection measures, did not publish any personally identifiable information, and restricted dataset access to users who agree to use it solely for scientific research. The release of our deception dataset aligns with international standards and industry best practices, positively impacting scientific research, technological progress, and public safety. We have added a new section in the revised manuscript to address these ethical considerations, ensuring a comprehensive and responsible approach to handling the deception detection dataset.
> 7. We have thoroughly revised the manuscript to improve the quality of writing and address all language issues, including specific errors such as "role.Although," "new multi-modal deception dataset," and reference formatting, as pointed out by the reviewers.

---

> > ### Author Response · Authors · 2024-11-18
> >
> > 8. The use of videos to induce emotions provides a controlled environment for eliciting specific emotional states, which is essential for scientific research. Although the social context of deception differs from watching a video, the emotional responses elicited by videos are ecologically valid, often depicting social interactions and emotional scenarios that mimic real-life situations. Videos can evoke a wide range of emotions, from basic to complex, which is important for studying how different emotions influence deceptive behavior and for training models to recognize a broad spectrum of emotional cues. The video-based emotion induction procedure provides a baseline for emotional responses that can be compared with those during the deception experiment, helping to identify discrepancies that might indicate deception, as individuals who are lying may struggle to regulate their emotions in a social context.
> > ﻿
> > We hope that these revisions address your concerns and strengthen the contribution of our paper. We are grateful for the opportunity to improve our work based on your valuable feedback. Thank you again for your comments and acceptance of our views.

---

> > > ### Comment · Reviewer_QeLm · 2024-11-22
> > > **Response**
> > >
> > > (1) If I understand correctly, the video emotion induction procedure is supposed to elicit emotional responses for which outward emotional display and internal affective state are congruent? Was this assumption checked / validated in some way?
> > > (4) Why is "relaxation" used as an "emotional baseline" when also "neutrality" is included in the set of emotions?
> > > (5) Simply performing the experiment several times and averaging does not imply anything about statistical significance. Instead a statistical test needs to be performed and clearly described (goal, assumptions, following common reporting standards).
> > > (8) I do not agree with the claim of ecological validity of video-based emotion induction, in particular when generalising to conversations. The at best marginal improvements for the model incorporating emotion information also raise doubts.
> > >
> > > Overall, I still think there are conceptual and methodological issues that speak against acceptance.

---

### Meta-Review · Area_Chair_bXw5 · 2024-12-17

**Metareview:**

This paper presents a new multimodal deception detection dataset along with personality and emotional characteristics. The dataset is comprehensive and can be useful to the research community, if open sourced.

However, the paper suffers from major methodological and conceptual issues, as pointed out by the reviewers. Unfortunately, the authors have not addressed these issues adequately and neither have they uploaded a revised version of the manuscript.

**Additional Comments On Reviewer Discussion:**

The paper suffers from major methodological and conceptual issues, as pointed out by the reviewers. Unfortunately, the authors have not addressed these issues adequately and neither have they uploaded a revised version of the manuscript.

---

### Decision · Program_Chairs · 2025-01-22

Reject